# A First Investigation of Agriculture Sector Perspectives on the Opportunities and Barriers for Agrivoltaics

**Alexis S. Pascaris** [1,*] **, Chelsea Schelly** [1] **and Joshua M. Pearce** [2,3,4]

1   Department of Social Sciences, Michigan Technological University, 1400 Townsend Drive, Houghton, MI 49931, USA; cschelly@mtu.edu

2   Department of Materials Science and Engineering, Michigan Technological University, 1400 Townsend Drive, Houghton, MI 49931, USA; pearce@mtu.edu

3   Department of Electrical & Computer Engineering, Michigan Technological University, 1400 Townsend Drive, Houghton, MI 49931, USA

4   School of Electrical Engineering, Aalto University, Aalto, 02150 Espoo, Finland

*   Correspondence: aspascar@mtu.edu; Tel.: 906-487-2113

**Abstract:** Agrivoltaic systems are a strategic and innovative approach to combine solar photovoltaic (PV)-based renewable energy generation with agricultural production. Recognizing the fundamental importance of farmer adoption in the successful diffusion of the agrivoltaic innovation, this study investigates agriculture sector experts' perceptions on the opportunities and barriers to dual land-use systems. Using in-depth, semistructured interviews, this study conducts a first study to identify challenges to farmer adoption of agrivoltaics and address them by responding to societal concerns. Results indicate that participants see potential benefits for themselves in combined solar and agriculture technology. The identified barriers to adoption of agrivoltaics, however, include: (i) desired certainty of long-term land productivity, (ii) market potential, (iii) just compensation and (iv) a need for predesigned system flexibility to accommodate different scales, types of operations, and changing farming practices. The identified concerns in this study can be used to refine the technology to increase adoption among farmers and to translate the potential of agrivoltaics to address the competition for land between solar PV and agriculture into changes in solar siting, farming practice, and land-use decision-making.

**Keywords:** agrivoltaics; solar energy; agriculture; energy innovation; technology adoption; photovoltaics

## 1. Introduction

The Intergovernmental Panel on Climate Change (IPCC) Carbon and Other Biogeochemical Cycles report [1] reveals the predominant sources of anthropogenic greenhouse gas (GHG) emissions are the use of fossil fuels as sources of energy and land use changes, particularly agriculture. Agrivoltaics, the strategic codevelopment of land for both solar photovoltaic (PV) energy production and agriculture, can meet growing demands for energy and food simultaneously while reducing fossil fuel consumption [2–4]. Integrated energy and food systems have the potential to increase global land productivity by 35–73% [2] and to minimize agricultural displacement for energy production [5–7]. Agrivoltaic systems are a strategic and innovative approach to combine renewable energy with agricultural production, effectively addressing the predominant sources of anthropogenic GHG emissions as identified by the IPCC.

The viability of emerging agrivoltaic innovation has been investigated in various contexts. In conjunction with solar PV, there are emu farms in Australia [8] as well as sheep grazing [6,9,10] and pollinator-friendly sites proliferating in the U.S. (e.g., [11]). There is also the potential to use agrivoltaics with rabbits [12] and aquaponics (aquavoltaics) [13]. Experimental agrivoltaic research is

occurring in diverse locations and climates. Examples include cultivation of corn and maize [14,15], lettuce [16,17], aloe vera [18], grapes [19], and wheat [20]. Mow [6] describes agrivoltaics as low-impact solar development that can alleviate agricultural displacement and assume varied designs: a solar-centric design that prioritizes solar output while growing low-lying vegetation; a vegetation-centric design that prioritizes crop production but incorporates solar panels and a colocation design that integrates both solar and agriculture for equal maximum dual output. Colocation designs have produced an estimated 3–8% per watt reduction in overall installation cost during site preparation due to cost reductions in land clearing and grubbing, soil stripping and compaction, grading and foundation for vertical supports, when compared to conventional solar industry development practices [6]. Further, Mavani et al. [4] found over a 30% increase in economic value for farms deploying such systems. Previous studies demonstrated that the dual-use of land for both PV and agriculture generates a mutually beneficial partnership that provides unique market opportunities to farmers and reduced operation and maintenance fees to solar developers, particularly in the case of grazing livestock [3,6,21–23].

The growing land footprint of solar PV presents social and spatial challenges, which are exacerbating the competition for land between agriculture versus energy production [5,23–25]. The U.S. Department of Energy Sunshot Vision Study forecasts that solar energy capacity will be nearly 329GW by 2030, which will necessitate approximately 1.8 million acres of land for ground-mounted systems [26]. Guerin [23] posits that the colocation of energy and agriculture will be stunted if there is absence of support from farmers and rural landowners, as the potential of agrivoltaic systems to address land-use competition will be contingent on farmer acceptance of agrivoltaics as a sociotechnological innovation. Brudermann et al. [27] found that PV adoption by farmers is primarily driven by environmental and economic considerations, which suggests factors that will be critical in agriculture sector decision-making concerning agrivoltaics.

Diffusion is a spatial and temporal phenomenon by which an innovation disseminates amongst adopters through a gradual process of filtering, tailoring and acceptance [28–30]. Rogers' [28] diffusion of innovations theory explains how and why some technological innovations are widely accepted while some are not, specifically referring to the adoption of an innovation by farmers over time in a rural diffusion model. The diffusion of innovations theory has been used to study diffusion of an innovation among physicians [31], among industrialized firms [32] and in terms of policy diffusion [33], among many other applications. Wilson & Grübler [34] applied the theory distinctly to energy innovations and described four phases of diffusion in which agrivoltaics can be categorized as existing in the first stage of an extended period of experimentation, learning, diversity of designs and small unit and industry-scale technologies. Grübler [30] warns that the existence of an innovation in itself does not promise proper diffusion, and while innovations have the capacity to induce change, it is the process of diffusion that realizes this potential as changes in social practice. By applying the diffusion of i theory to the agrivoltaic innovation, this study seeks to offer insight into potential refinements to the innovation of agrivoltaics in terms of its social acceptance to enable continued diffusion. This study uses Rogers' theory [28] as a practical framework for informing the diffusion of agrivoltaic innovation to discern the future potential and challenges for this technology to diffuse sufficiently to address energy and agricultural demands sustainably. While the technical viability of colocating solar PV and agriculture has been demonstrated [2,3,16,17], research in this field is incomplete with regard to placing the innovation within a social context to determine barriers to diffusion as perceived by industry experts.

Recognizing the fundamental importance of farmer adoption in the successful diffusion of agrivoltaics, this study investigates agriculture sector experts' perceptions on the opportunities and barriers to dual land-use agrivoltaic systems. Using in-depth, semistructured interviews, this study seeks to further the potential of agrivoltaics by identifying challenges to farmer adoption in an effort to address them by responding to societal concerns. In the following sections, the results are discussed, and conclusions are drawn on barriers to be overcome for agrivoltaic diffusion as identified by industry experts. The organization of the results and discussion are based on concepts from the diffusion of innovations theory [28], with a focus on relevant innovation characteristics (observability, relative

advantage and compatibility), stages of the adoption process and categories of adopters. Finally, the implications of these findings for the future development of agrivoltaics and farmer adoption are considered.

## 2. Materials and Methods

This study investigates agriculture sector experts' perceptions of the opportunities and barriers to agrivoltaics using in-depth, semistructured interviews. Interview methodology is exploratory by nature and, most appropriately, collects and analyzes data about perceptions, opinions and attitudes of people [35]. Aimed at providing an inclusive and nuanced perspective of the phenomenon under study, interviews were employed to directly engage relevant informants related to agriculture and agrivoltaics.

Prior to commencement, this research obtained approval from Michigan Technological University's Institutional Human Subjects Review Board (code: 1524021-1) to ensure compliance with institutional ethics in human subjects research. The initial interview protocol can be found in Appendix A. Email was used to introduce the agrivoltaic concept and the study while inviting prospective participants to video conferencing discussions, which resulted in 10 online interviews lasting between 30 to 90 min. All participants provided informed consent for the recording of conversations, which were anonymized for the protection of their privacy. Data collection occurred between February and July 2020 until saturation was attained, known as the point when no new additional insight is derived from conversations with participants and stabilization of data patterns occur [36,37].

A total of 10 interviews were conducted with 11 agriculture sector professionals (one interview engaged two individuals simultaneously), including livestock and crop farmers, solar grazers (individuals who graze their livestock underneath solar panels) and an agriculture policy expert. Sampling for logical representativeness, variance, diversity, and relevance to agriculture, participants were pursued based on their potential to provide insight into the opportunities and barriers to agrivoltaics because they have direct experience in the agricultural sector. Both theoretical and snowball sampling methods are nonprobability techniques that were employed to construct a sample capable of representing a wide range of perceptions. Theoretical sampling intentionally captures individuals with certain characteristics [38,39], whereas snowball sampling progressively follows a chain of referrals from study participants to other potential contributors [40,41]. Table 1 details the sample of participants that was generated using these sampling methods, ranging in profession, geographic location and gender. While credible and valuable, samples constructed through nonprobability sampling do not lend themselves to generalization [42], nor are the findings generated through interview methodology suitable for statistical generalization or analysis. However, all of the themes discussed as findings were raised by the majority of participants and identify the primary opportunities and barriers to agrivoltaics according to this sample but cannot be quantified or suggested to represent a broader population. Therefore, the findings are not discussed quantitatively to steer clear from suggesting these results are statistically generalizable to the entire agriculture sector.

**Table 1.** Interview Participant Characteristics.

| Profession | Geographic Region (United States) | Gender |
|---|---|---|
| Livestock farmer: 5 | North East: 4 | |
| Crop farmer: 1 | South East: 1 | Male: 5 |
| Solar grazer: 4 | Midwest: 5 | Female: 6 |
| Policy: 1 | South West: 1 | |

Drawing from grounded theory methodology [41,43], data collection and data analysis occurred in parallel to strategically shape subsequent inquiry. Responses that emerged in initial interviews instructed the development of ensuing questions, allowing for gradual pursuit and refinement of relevant issues. Interview themes were generally organized around: (1) the participants' experience in agriculture and details of their current operation; (2) experience with and perceptions of agrivoltaics (e.g., attitudes,

opinions, perceived opportunities and barriers); (3) willingness to engage in an agrivoltaic project (e.g., perceived benefits and challenges). Interview protocol matured over time to explicate what agriculture sector professionals perceived as relevant opportunities and barriers to agrivoltaic development.

All interviews were recorded, manually transcribed and analyzed using the qualitative data analysis program NVivo 12 Pro (QSR International, Melbourne, Australia) [44]. Data were studied on a line-by-line basis using a series of coding and analytic induction to explore relationships, patterns and processes. Line-by-line coding is the fundamental step in interview analysis that moves beyond concrete statements to make analytic interpretations [41]. Coding in grounded theory methodology helps anchor analysis to participants' perspectives, explore nuances of meaning, identify implicit and explicit issues, as well as cluster similarities and observe differences among responses [41]. As outlined by Znaniecki [45] and Robinson [46], analytic induction involves identifying patterns, themes and categories in qualitative data in preparation for comparison amongst the varied findings. Employing rigorous, iterative and comparative grounded theory techniques, analysis of these data has captured and condensed the most relevant opportunities and barriers to agrivoltaics according to this sample of agriculture sector professionals.

## 3. Results

This section organizes findings based on frequency and expressed magnitude of the barriers and opportunities to agrivoltaics as defined by study participants. Both direct quotations (italicized) and analysis of results are presented jointly. Sections 3.2 and 3.3 are aligned with three of the five innovation characteristics defined by Rogers' diffusion of innovations theory [28] (observability, relative advantage and compatibility), which were identified by participants as the most critical when considering the adoption of agrivoltaic technology. These results offer insights into the main challenges to farmer adoption of agrivoltaics and suggest opportunities for interested stakeholders to further diffuse this innovation. A discussion considering the implications of these results is followed in Sections 4 and 5.

### 3.1. Long-Term Land Productivity and Planning

The underlying fundamental challenge of agrivoltaic systems, as perceived by participants, concerns long-term land viability. Land viability is intrinsically proportionate to the livelihood of agriculturalists, as farmers explained that the quality of their land is of critical importance and cannot be compromised. Interviews with farmers revealed their temporal approach to decision-making as they prioritize the protection of long-term land viability above all. One farmer expressed this concern when considering the use of an agrivoltaic system:

> *I'm concerned too, if you're pouring a bunch of concrete and putting in permanent structures, what does this look like in the end of 20 or 30 years?*

Encompassed within concerns of long-term land viability are more nuanced challenges related to land productivity in the presence of permanent solar panel structures. Participants explained that in order to maintain their agricultural land status and thrive in their farming venture, land must stay actively agricultural. The challenge that permanent solar structures could potentially impose on land productivity was unsettling:

> *Given the permanency of all of the solar panels and the permanency of the size of the plot, maintaining it to be continually productive for the animals would be a challenge. One of the challenges that I foresee is learning how to get the production that you want navigating around all of those structures.*

When considering an agrivoltaic system, participants' concerns were largely technical and economic in nature, reflecting their dependence on land productivity. Considerations about long-term land use and farmland preservation constituted the basis for decision-making, suggesting that anything that jeopardizes land viability will not be tolerated by farmers. Thinking beyond protecting the soil

itself, various participants expressed potential opportunities that agrivoltaic systems could bring to agriculturalists:

> *When we talk about farmland preservation, it's not just about preserving the physical ground, it's also about preserving the viability of the farm. If a farmer is going to go under because of lack of revenue, why wouldn't you want them to open up an additional revenue stream to be able to actually preserve that land?*

> *There's going to be ground that goes into the solar panels and I think the idea that here you can integrate mixed-use with this makes a lot of sense. I think you have to have the right farmers and the right producers that are committed to making some of these things work.*

Participants explained that long-term land viability and productivity implies required long-term planning. When discussing the prospect of engaging in an agrivoltaic project, participants proposed that incorporating some type of land-use agreement or long-term plan would relieve concerns around the future of their farm. Providing certainty of farmland preservation surfaced as a recurring consideration of agrivoltaic adoption, as articulated by one participant:

> *Restoring the land back to what it was having the right land agreements to where when that lease is up, they have to return it to prelease form.*

To address the need for long-term planning and prioritization of agricultural interests, agrivoltaic project contracts are widely used by current stakeholders. As described by interviewees who identify as solar grazers, agrivoltaic contracts provide certainty and prevent against loss for both parties involved. The temporal concerns of agriculturalists with regards to long-term land viability can be reassured by agreement and engagement on both sides, as a solar grazer explains:

> *You can't have any business planning when you have that degree of uncertainty. So, it was getting people to have contracts. What the contract did is give certainty to both sides. It meant the farmers could plan their businesses, because there is a whole bunch of this remote targeted grazing, there's tons of mechanics, tons of money, staffing, and planning around breeding schedules, you name it. And then on the other side you got people wanting to make sure that the insurance is okay, and that their wiring is going to be okay, and how they'll interface with all their service work, the whole picture. I just knew the contract was the first key to the puzzle.*

> *If you don't have a real contract and if you don't have someone really interested engaging in a 10-year kind of way on both sides, the whole thing is not going to work.*

The majority of participants communicated that to the extent that the solar infrastructure of an agrivoltaic project does not threaten long-term land productivity, there are opportunities for increased revenue to farmers and mutually beneficial land-use agreements. These interviews reveal that addressing concerns about the viability of land after project decommissioning and protecting the livelihoods of farmers will involve long-term planning and partnership between agriculture and solar industries. The establishment of agrivoltaic contracts has proven valuable to current solar grazers and provides a direct way to alleviate uncertainties in land-use planning.

### 3.2. Market (Un)certainty and Observability of Benefits

When considering barriers to farmer adoption of agrivoltaics, economic concerns were raised by participants only second to concerns described above regarding long-term planning for technical considerations. At a basic level, farming is a business, and is thus accompanied by a set of risks, uncertainties and investments. Participants explained that risk is especially unwelcome in the business of farming and that certainty in productivity and security in investment are vital. One participant articulated that the market unknowns are potentially more critical than the technical unknowns of agrivoltaics:

*There's a lot of unknowns for the producer in this as well. Having established markets, alleviating some of the unknowns and the risks are probably as much of a piece of this as anything. So, sketching out the long-term financial return of like, "Here's what these markets look like for livestock production." And what the guaranteed revenue is for solar panels, for instance. In terms of just making it happen out there in the field, there's some requirements to make that happen, but they aren't insurmountable, I wouldn't imagine.*

Others stressed the need for a secure market for an agrivoltaic system to be successful:

*You would probably want to package it more as, "Do we have a food and farm system in place that allows somebody to have solar and grow these crops that are tolerant to that condition?" And then importantly, "Do we have a market to send that stuff to?" Because then all of a sudden it becomes this closed loop, kind of circular economy feel to it. But without that end market side of it, I think people would say, "That's great if you want to grow that stuff."*

*As long as the market is there, I would think a lot of these things could work.*

As business owners, considerations of financial return and security in the marketplace are at the forefront of decision-making for farmers. For the majority of participants, the agrivoltaic innovation is unfamiliar and imposes constraints on business planning borne of unknowns and uncertainties. Building flexibility into the system to accommodate for changes in market conditions and farming practice could potentially alleviate some of the concern of uncertainty, as explained:

*If we're looking at a 25-year kind of investment with the solar panels and when you're talking about integrating them within the livestock species too, that market for livestock might look totally different within 10 years. So, implementing some flexibility there that if we're not going to run rabbits, maybe we're running something else in there in 20 years. But having some flexibility in the system that you could respond to the livestock markets in there as well, I think is important.*

Flexibility and adaptation to changing market conditions emerged as key elements to be incorporated into planning for an agrivoltaic system, highlighting again the temporal component to farmer decision-making and identifying concerns to be addressed for successful adoption. While the future unknowns of market acceptance of a product are difficult to ascertain, participants suggested that integrating flexibility into system design would reduce financial unease.

Coupled with concerns of a stable and reliable market for their product, were expectations for just compensation and tangible benefits from participation in an agrivoltaic project. When considering the adoption of the agrivoltaic innovation, participants also questioned if such an endeavor would be justified in terms of monetary gains. Participants perceived the adoption of such technology as an increased labor commitment and thus expected to reasonably gain from it. When asked if they would engage in an agrivoltaic project, one participant answered:

*Essentially, they would have to pay me if they wanted me to be there because it's so much work to remediate soil and bring it up to a productive level, especially if this has been formally row cropped conventionally. So, it would really depend on what it had been earlier, how much I trusted the people who were starting this operation, and how much I felt that there would be ease of incorporating it into my schedule. I also think that it's not free pasture, you know what I mean? Even if they didn't charge me a single thing, there would be a lot of investment. So, I'd be going for like- I don't even know- I almost want to see like co-ownership, we own this land together, you get the profits from the solar and I get whatever everything else is. Or putting the solar panels on my own farm and then I get the revenue from the solar panels.*

When judging the adoption of agrivoltaic innovation, participants expressed critical valuations of its worth and asserted that observable and substantial benefits would have to be derived in order

for them to commit. Of the 10 farmers interviewed, four were already engaging with the technology and five others said they would get involved if they would derive more benefit than cost from it. Thus, the vast majority (nine of 10) of the farmers interviewed were open to using or already using agrivoltaics. Improving the agrivoltaic innovation to increase diffusion to these interested farmers will require establishment of just compensation for farmers, as explained by two solar grazers:

> *The biggest misconception to clear up immediately when people start thinking about this is that it can be anything like free grass. Because there's so much commitment on my end, and the cost of setting up all that equipment is very high. The time and labor of going there and servicing the sheep is a big commitment.*

> *I'm really trying to get out of is the idea that the farmer should be doing all this work for free. The solar firms are making—maybe not tons of money—but reasonable amounts of money off these investments. For them, they need to know that the performance guarantee is there, the sun has to shine on their panels, there shouldn't be interference with that. They need that steady assurance. And the farmers need to get paid for recognizing that there is a performance guarantee to meet.*

Participants explained that their willingness to be involved with the agrivoltaic innovation would be contingent on the near-term observability of direct benefits to them and the long-term certainty and security in the marketplace for their product. Observability is an innovation characteristic explained by Rogers (1962) that concerns the degree to which the results of an innovation are visible to potential adopters. When assessing their potential adoption of agrivoltaics, agriculture sector experts framed their considerations in terms of direct and tangible benefits, suggesting that observability of benefits is a characteristic of the agrivoltaic innovation that is of decisive importance to adopters. As discussed by participants in Section 3.1, agrivoltaic contracts are currently recognizing the rights and duties of involved parties, and provide opportunity to establish legitimate, mutually beneficial partnerships. With nine of 10 farmers inclined to partake in an agrivoltaic partnership, the above concerns about economic uncertainty and gains are active considerations for all involved stakeholders in project development.

Relative Advantage

The degree to which agrivoltaics are perceived by participants to be advantageous to current practice was identified as important when considering adoption. While participants expressed that financial compensation for farmers is both necessary and attractive, they also spoke of other benefits they anticipate as a result of engaging with the agrivoltaic technology. Participants discussed potential marketing advantages:

> *It's got a great story; it's got a wonderful marketing edge from that perspective. So, your advantage is a great story to tell from a marketing standpoint.*

> *I think that's where you have a very unfair advantage for whoever would be doing this rabbit production, you might be getting paid for land maintenance and then have rabbits for free. So, your profitability could be way up or your price could be way lower because you wouldn't have land expenses. There's a lot of opportunity to create some advantage from a production standpoint. From that perspective they may sell better or have an [edge] in the marketplace because of that aspect.*

Another participant expressed other technical synergies when grazing animals underneath solar panels:

> *I think it sounds like a great idea. It sounds like a great way to maintain, and not have to mow. I can see the panels providing shade and protection from the rain in a way that seems very valuable.*

Perceiving a multitude of potential benefits, participants speculated how the adoption of the agrivoltaic innovation could provide them benefits and competitive advantages in the marketplace. Foreseeing a unique opportunity to derive a revenue stream from land maintenance, some participants postulated that there were economic gains associated with combined solar and agriculture systems. Rogers' (1962) innovation characteristic, relative advantage, explains that innovations that are perceived to be superior to business as usual have higher potential for adoption. Participants described the relative advantage of agrivoltaics worthwhile, and thus identified this innovation characteristic as critical when considering the adoption of the innovation, suggesting that if an agrivoltaic system could provide an advantage to a farmer, the likelihood of adoption would be greater.

*3.3. Compatibility with Current Practice*

A considerable opportunity for farmers in agrivoltaic projects is the potential for integration of the innovation into their current practice. Participants expressed disinterest in increased complications in their business, and rather actively seek ways to reduce labor through harnessing the synergies of innovative practices. The ease of integration and compatibility of solar with current production was frequently considered amongst participants, highlighting the opportunity to plan overlapping operations to increase farmer acceptance. The attractiveness of agrivoltaic integration was explained by two participants:

> *Most of my exposure to this is from sheep, and I think that it's a great idea. For my own particular system, it would definitely reduce the amount of labor for one aspect of the system, which is moving the fencing. So, I'm all for it. I think it'd be a really nice mesh.*

> *Alternative energy is expensive to people like us. But it's something that I guess, if it could be integrated into something I'm already doing and could potentially help protect the animals, or do whatever, and then also run the homestead, it's just another perk of having something like that. It's another reason to have it besides just having the electricity.*

As elucidated by participants, compatibility of the agrivoltaic innovation with current practice could reduce labor and create an incentive to engage in the technology. When considering the value of agrivoltaics to them personally, farmers offered calculated and context-dependent perspectives, making judgments on the benefits in terms of their own operation rather than speaking generally about dual-use solar systems. Speaking from a place of personal considerations and interests, participants revealed that there is a context-dependent nature of success for agrivoltaic projects. Reflecting their own practices, one participant stated:

> *I've also heard them say in meetings the fact that we're going to farm soybeans underneath solar panels, which is just asinine. Like, it's not going to happen. The size of our equipment doesn't permit that kind of thing. Putting livestock under, kind of a grazing operation, seems to make sense.*

Compatibility with current practice not only includes size of equipment, but also scale of the farming operation, as explained by one participant:

> *The work that would be involved with that, I think, or potentially having to hire someone to manage them, it would decrease our profit so much that it wouldn't make sense. I could see how that would be to someone's benefit though, but not at our scale.*

To justify the labor involved in engaging in an agrivoltaic project, farmers evaluated their own enterprise by mentally applying the innovation and determining the potential compatibilities. As suggested by participants, the benefits of agrivoltaics are noteworthy, but will only be fully realized if there is ease of integration into their current farming practice. Compatibility is an innovation characteristic defined by Rogers (1962) that explains the degree to which an innovation is perceived to be consistent with needs, norms and sociocultural values is decisive to potential adopters. The theme

of compatibility among most participants was viewed as an opportunity rather than a barrier for agrivoltaics, suggesting that the innovation's context-dependent nature provides flexibility and potential to leverage the solar system to derive synergistic benefits to compliment current farming practices.

## 4. Discussion: The Opportunities & Barriers for Agrivoltaic Diffusion

This research provides insight from the agricultural sector into the challenges and opportunities for farmer adoption of the agrivoltaic innovation. Results indicate that participants see potential benefits for themselves in combined solar and agriculture technology and identify barriers to adoption including desired certainty of long-term land productivity, market potential and just compensation, as well as the need for predesigned system flexibility to accommodate different scales of operation and adjustment to changing farming practice. The findings suggest that these barriers to adoption are not insurmountable and can be sufficiently addressed through prudent planning and mutually beneficial land agreements between solar and agriculture sector actors. Table 2 below organizes the identified barriers and opportunities to address them. All of the participants of this study assented to agrivoltaics as a synergistic and innovative approach to combined land-uses, while nine of the 10 participants who are currently active farmers stated they would engage in the use of a dual-use system given the discussed concerns are considered (four of the nine already are). Interviews with industry professionals informed the current state of diffusion of the agrivoltaic innovation and identified opportunities to further stimulate farmer adoption of the technology. These findings may be used to translate the potential of agrivoltaics to address the competition for land between solar PV and agriculture into changes in solar siting, farming practice and land-use decision-making.

**Table 2.** Barriers, opportunities, and directions for future work regarding the diffusion of agrivoltaics.

| Barrier | Opportunity | Future Work |
|---|---|---|
| End-of-life impacts from solar infrastructure | -Driven piles (constructed of galvanized steel I-beams, channel-shaped steel or posts), helical piles (galvanized steel posts with split discs welded to the bottom at an angle) and ground screws (galvanized steel posts with welded or machined threads) can be removed and recycled [47,48].<br>-Photovoltaic (PV) racking can be put on removeable ballasted foundations or skids of precast or poured-in-place concrete ballasts to minimize land disturbances [47].<br>-Impacts from modules such as leaching of trace metals [49–51] and compromised future agricultural productivity [52] have been proven highly unlikely.<br>-Contracted agreements that establish plans to return land back to prelease form after decommissioning of solar system. | -Empirical research investigating the magnitude of long-term impacts of solar infrastructure on land (e.g., [53]), soil, and pasture-grass productivity. |
| Permanent structures interfering with agricultural production and future farming practice | -A variety of plants have proven to maintain higher soil moisture, greater water efficiency, and experience increase in late season biomass underneath PV panels [54].<br>-Improvements in water productivity and additional shading are projected to increase crop production in arid regions experiencing climate change [55].<br>-Semitransparent PV [56] (Thompson et al., 2020) or vertical bifacial PV [57].<br>-Raised racking systems provide clearance for agricultural equipment, which could allow for nearly any crop to be used in agrivoltaic production [58].<br>-Design flexible open source racking systems [59,60] that have adjustable panel height, tilt angle and spacing [61], as well as a combination of permanent and portable fencing.<br>-East-west tracking array configurations allow optimal conditions for plant growth when compared to conventional south-facing designs [62]. | -Empirical research aimed at understanding the implications of solar PV infrastructure on perennial pasture grass maintenance.<br>-Optimized agrivoltaic PV<br>-Cost-benefit analysis of open source PV racking systems designed with adjustable panel height, tilt angle and spacing.<br>-Cost-benefit analysis of permanent and portable fencing for animal grazing agrivoltaics. |

**Table 2.** *Cont.*

| Barrier | Opportunity | Future Work |
|---|---|---|
| Uncertainties in operation and business planning | -Legitimate partnerships and contracts that establish up-front costs and compensation for both parties<br>-Local government policy aimed at supporting development of solar PV [63,64]<br>-Education and outreach from PV industry to farming industry to reduce barriers to knowledge and increase trust. | -Policy research focused on market mechanisms to incentivize agrivoltaic systems for both solar and agriculture sector.<br>-Increased efforts from university extension programs to increase information sharing and partnership between energy and agriculture. |

### 4.1. Diffusing the Agrivoltaic Innovation—Where Are We Now?

The diffusion of innovations theory [28] identifies five stages in the process of technology adoption. Participants of this study predominantly fell into the decision or evaluation stage of adoption, which is understood as the stage in which an individual mentally applies an innovation to their present and perceived future circumstances to arrive at a decision to try it or not. Beyond the initial knowledge or interest stages of Rogers' adoption model [28], the majority of participants (six of 11) considered their potential adoption of agrivoltaics beneficial but dependent on factors related to context. Speaking from a place of receptivity, these participants saw value in the innovation and felt inclined to engage with it, while voicing a few concerns about compatibility with their practice and uncertainties about long-term land productivity. Four of the 11 participants were already functioning in the confirmation or adoption stage of the adoption process, making full use of the innovation. Based on these findings, it is observed that the current state of the diffusion of agrivoltaics is advancing towards wider implementation and has surpassed initial phases of information gathering and persuasion. Participants in the decision or evaluation stage of adoption identified barriers to their engagement with agrivoltaics, giving interested stakeholders the ability to directly respond to these concerns by improving the technology to enable further diffusion.

Further, most participants of this study were early majority adopters, characterized by wanting proven and reliable applications, reference from trusted peers and being prudent in financial risk and uncertainty. Rogers [28] asserts that an innovation must meet the needs of all categories of adopters, making clear in the context of agrivoltaic adoption where efforts should be focused to successfully move early majority adopters into acceptance of the innovation. Technological diffusion is a process of filtering, tailoring and accepting [30], and the identified concerns of the agriculture sector professionals in this study can be used to tailor or refine the technology to increase adoption among farmers. The following section will elaborate upon the critical characteristics of agrivoltaic systems as identified by participants and suggest recommendations for improvement with the intention of facilitating accelerated diffusion.

### 4.2. Diffusing the Agrivoltaic Innovation—What Needs to Happen?

Rogers [28] posited that there are five distinct innovation characteristics that help explain why some innovations are widely accepted and some are not. Understanding the characteristics of the agrivoltaic innovation is valuable for interested stakeholders when assessing areas for improvement and pursuing further acceptance of the technology. The results of this study identify the most critical characteristics of agrivoltaics and point to opportunities to directly respond to farmers concerns.

Of these five characteristics, observability of benefits, relative advantage and compatibility with current practice were identified by participants as the most critical when considering their personal adoption of the agrivoltaic technology. What this means for further diffusion is that the solar industry actors involved in the development of agrivoltaic systems must devise mutually beneficial land agreements with farmers that establish compensation for their labor, articulate plans for land restoration after the decommissioning of the system and be sensitive to contextual differences among agriculturalists by designing a system that is flexible enough to meet the needs of the current and

future users. Participants in this study saw immediate value in personal adoption of the technology but sought long-term security in terms of farmland preservation and financial return.

There are a handful of practical actions to be taken to enable further diffusion of agrivoltaics. Table 2 presents a summary of the identified barriers, existing opportunities to overcome them and directions for future work. First, the establishment of agrivoltaic contracts has proven valuable to current solar grazers. Robust and forward-thinking land use agreements will provide a direct way to alleviate uncertainties in land-use planning and secure compensation for farmer's labor. Second, system designers need to integrate flexibility in design by accommodating current land practices and allowing for future changes. Concerns about market uncertainty and rigid systems can be addressed by crafting a combined solar and agricultural project that is adaptable to changing market and farming conditions. Third, agrivoltaics systems should be designed with compatibility in mind. By strategically harnessing the synergy of compatibility with current practice, these results suggest that farmers would be more inclined to engage with a project if it generated advantages in their operation. Being sensible in scaling a system to current practice, rather than creating increased labor burden on farmers, will increase the likelihood of their participation with the technology.

The potential for increased utilization of the agrivoltaic technology is ripe. While previous research has demonstrated its technical viability, this study recognizes that technology innovations exist within a social context and thus depend upon social acceptance and adoption. It is concluded that continued farmer adoption of agrivoltaics is likely, yet contingent on observable benefits in farming practice and assurance of financial gain. Future research should investigate how perceptions vary across geographic regions and agriculture professions (i.e., animal versus crop farming) to study the unique opportunities and barriers for agrivoltaics in the context of local climate and agricultural practice. Increased education and outreach concerning the end-of-life impacts, negligible effects of solar PV on agricultural productivity and potential for agrivoltaic systems to protect crop production during climate change, is necessary to inform and stimulate further farmer adoption. Empirical experimental research should investigate the long-term impacts of solar PV infrastructure on perennial pasture grasses to better understand the possible effects of agrivoltaic systems on future grazing productivity. Economic cost-benefit analysis will be valuable for quantifying the potential cost disadvantages of designing flexible PV arrays that can be adjusted to accommodate different panel heights and spacing requirements. Future policy research can investigate the role of market mechanisms, such as incentives, in prompting further development of agrivoltaics. Based on these findings, policy makers should consider implementing financial instruments that stimulate both solar and agriculture sector adoption of the technology, while building flexibility into such policies to allow diverse, innovative and contextually appropriate system designs. To do this, agrivoltaic proponents can model their efforts on the successful diffusion of wind farm/solar farm integration that focuses on local support [65,66]. Previous research examining diffusion of solar as an innovation among residential adopters highlighted the role of communities of information sharing for promoting adoption [67]. The study presented here is unique in examining the diffusion of agrivoltaic solar innovation as a community level consideration, but also demonstrates how diffusion of innovation can occur within a social context. Moving forward, placing the agrivoltaic technology in a social context will be essential to identify the barriers to its diffusion and will offer relevant solutions to increase its adoption.

## 5. Conclusions

Agrivoltaic systems are a strategic and innovative approach to combine renewable energy with agricultural production. Recognizing the fundamental importance of farmer adoption in the successful diffusion of agrivoltaics, this study investigates agriculture sector experts' perceptions on the opportunities and barriers to dual land-use systems. Results indicate that participants saw potential benefits for themselves in combined solar and agriculture technology and identified barriers to adoption including desired certainty of long-term land productivity, market potential and just compensation, as well as the need for predesigned system flexibility to accommodate different scales

and types of operations and adjustment to changing farming practice. The identified concerns of the agriculture sector professionals in this study can be used to refine the technology to increase adoption among farmers and to translate the potential of agrivoltaics to address the competition for land between solar PV and agriculture into changes in solar siting, farming practice and land-use decision-making. Ultimately, building integrated energy and food systems can increase global land productivity, minimize agricultural displacement and reduce greenhouse gas emissions from fossil fuels. Informed and concerted efforts at enabling further diffusion of this innovation are imperative for meeting growing demands for energy and food simultaneously.

**Author Contributions:** Conceptualization, C.S. and J.M.P.; methodology, A.S.P. and C.S.; validation, A.S.P. and C.S.; formal analysis, A.S.P. and C.S.; investigation, A.S.P.; resources, C.S. and J.M.P.; data curation, A.S.P.; writing—original draft preparation, A.S.P.; writing—review and editing, A.S.P., C.S. and J.M.P.; supervision, C.S. and J.M.P.; project administration, J.M.P.; funding acquisition, C.S. and J.M.P. All authors have read and agreed to the published version of the manuscript.

**Funding:** This material is based upon work supported by the U.S. Department of Energy's Office of Energy Efficiency and Renewable Energy (EERE) under the Solar Energy Technology Office Award Number DE-EE0008990 and the Witte Endowment.

**Conflicts of Interest:** The authors declare no conflict of interest. The funders had no role in the design of the study; in the collection, analyses, or interpretation of data; in the writing of the manuscript, or in the decision to publish the results.

**Disclaimer:** This report was prepared as an account of work sponsored by an agency of the United States Government. Neither the United States Government nor any agency thereof, nor any of their employees, makes any warranty, express or implied, or assumes any legal liability or responsibility for the accuracy, completeness, or usefulness of any information, apparatus, product, or process disclosed, or represents that its use would not infringe privately owned rights. Reference herein to any specific commercial product, process, or service by trade name, trademark, manufacturer, or otherwise does not necessarily constitute or imply its endorsement, recommendation, or favoring by the United States Government or any agency thereof. The views and opinions of authors expressed herein do not necessarily state or reflect those of the United States Government or any agency thereof.

## Appendix A

Initial interview protocol as approved by IRB

1. Please tell me about your experience as a farmer.

    a. What is your geographic location?
    b. How long have you been doing it?

2. Who [markets, restaurants] are your biggest customers?

    a. How do you go about opening new accounts with potential customers?
    b. What is your greatest barrier to gaining access to new markets/customers?

3. How large is your operation? Would you consider it small-medium-large?
4. Are you familiar with both crop and animal farmers that incorporate solar panels on their land?

    a. If so, what are your thoughts on this?

5. Would you ever consider embracing the mixed-use of solar on your farm to harness co-benefits of solar energy generation and agricultural production?

    a. If so, why?

        i. What is your minimum acceptable rate of return?
    b. It not, why?

        i. What type of barriers are there?

6.  Would you consider renting land on a prefenced solar-farm meant for agricultural production?

    a.  If so, why?

        i.   What is your minimum acceptable rate of return?

    b.  It not, why?

        i.   What type of barriers are there?

7.  What is needed to make a mixed-use solar farm more attractive to you?

8.  A new study that is sponsored by the D.O.E. has shown an opportunity to incorporate rabbit farming with solar photovoltaic farms that make electricity. This study has shown substantial economic opportunity from this mixed-use scheme: upwards of 24% increase in site revenue. Now I would like to ask you specifically about mixed-use solar involving farmed meat rabbits.

    a.  What do you think are the biggest opportunities for this kind of mixed-use solar development?

    b.  What do you think are the biggest barriers for this kind of mixed -use solar development?

9.  Do you anticipate solar farm pasture-raised livestock selling for a premium or increasing sales?

10. Is there anything else you would like to tell me about your perspectives of mixed-use solar PV development?

11. Do you have suggestions of other experienced farmers I should speak with?

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
