# Peer review of "A First Investigation of Agriculture Sector Perspectives on the Opportunities and Barriers for Agrivoltaics"

_agronomy, doi:10.3390/agronomy10121885_

Round 1

Reviewer 1 Report

The research topic is of great interest. However, the extremely small sample of interviewees does not allow us to consider the results sufficiently representative.
Before a possible resubmission of the article it is recommended to considerably enlarge the sample.

Line 86: Is it true that it has been amply demonstrated?
E.g. in ref. 2, authors state: "There is a need to validate the hypotheses included in our models and provide a proof of the concept by monitoring prototypes of agrivoltaic systems."
In ref. 3: "It is clear, further work is warranted in this area and that the outputs for different crops and geographic areas should be explored to ascertain the potential of agrivoltaic farming throughout the globe."
In ref. 7 the study shows in particular the benefits from the point of view of energy production, without addressing the impact on agricultural crops.
Even the study in ref. 14 considers above all energy production.

The term "agriculture industry" used in several parts in the text is misleading. In many countries this term indicates the primary processing industries of agricultural products. Better to simply use "agriculture".

As the authors themselves affirm, a sample of 11 respondents cannot be considered representative and the results cannot be generalized. This would suggest at least modifying the title of the manuscript focusing it on a "first survey" or a "local survey".

Table 1. geographical region?

  What do you want to do ? New mailCopy   What do you want to do ? New mailCopy

Author Response

  1. The research topic is of great interest. However, the extremely small sample of interviewees does not allow us to consider the results sufficiently representative.
    Before a possible resubmission of the article it is recommended to considerably enlarge the sample.

It is true that the sample size and sampling method do not lend these findings to statistical generalizability or representation of a broader population. However, statistical generalizability is not the aim of interview methodology, which rather seeks theoretical generalizability. Because this study is provisional, external validity of case selection was sought in terms of logical representativeness, diversity, and relevance to the purpose of this study. Rather than increasing the sample size to meet the requirements of statistical generalizability, an explanation is provided (see below) regarding the conventional use of interview samples to generalize theoretical propositions in social science research.

Fixed: {While credible and valuable, samples constructed through non-probability sampling do not lend themselves to generalization [42] nor are the findings generated through interview methodology suitable for statistical generalization or analysis. However, all of the themes discussed as findings were raised by the majority of participants and identify the primary opportunities and barriers to agrivoltaics according to this sample but cannot be quantified or suggested to represent a broader population. Therefore, the findings are not discussed quantitatively to steer clear from suggesting these results are statistically generalizable to the entire agriculture industry.} Line #130-136

  1. Line 86: Is it true that it has been amply demonstrated?

E.g. in ref. 2, authors state: "There is a need to validate the hypotheses included in our models and provide a proof of the concept by monitoring prototypes of agrivoltaic systems."
In ref. 3: "It is clear, further work is warranted in this area and that the outputs for different crops and geographic areas should be explored to ascertain the potential of agrivoltaic farming throughout the globe."
In ref. 7 the study shows in particular the benefits from the point of view of energy production, without addressing the impact on agricultural crops.
Even the study in ref. 14 considers above all energy production.

Reference to articles #7 and #14 have been removed and replaced by reference to two research articles confirming improved crop yield [16], land use efficiency and water productivity [17] as a result of co-locating solar PV and agriculture.

Fixed: {While the technical viability of co-locating solar PV and agriculture has been demonstrated [2, 3, 16, 17], research in this field is incomplete with regard to placing the innovation within a social context to determine barriers to diffusion as perceived by industry experts.} Line #87-88

  1. The term "agriculture industry" used in several parts in the text is misleading. In many countries this term indicates the primary processing industries of agricultural products. Better to simply use "agriculture".

Fixed: {We have removed industry and either used agriculture or agriculture sector to make it clearer for the readers.}

  1. As the authors themselves affirm, a sample of 11 respondents cannot be considered representative and the results cannot be generalized. This would suggest at least modifying the title of the manuscript focusing it on a "first survey" or a "local survey".

Fixed: {Title of the manuscript modified as “A First Investigation of Agriculture Sector Perspectives on the Opportunities & Barriers for Agrivoltaics”}

  1. Table 1. geographical region?

Fixed: {Geographic Region of the *United States* has been clarified} Line #159

Reviewer 2 Report

The paper presents some interesting information on farmer’s perception to the ‘Agrivoltaics’ concept. I liked the how this was addressed in the paper. However, there are a few things that the authors need to take a look at:

1.There are no 'References' given in the article at the end.

2.Please remove the last line from the Abstract: "It is concluded that building integrated energy and food systems can increase global land productivity, minimize agricultural displacement, and reduce carbon emissions from fossil fuels." I don't think this is the conclusion from this paper.

3.It looks like the number of interview samples are very low. In some geographic locations, it has only one participant. There is also one crop farmer. Why not keep the number of samples the same for each profession and geographic region? Then we could have seen differences in perceptions across profession and geographic regions. Do all the participants have an agrivoltaic system or have experience with one ?

4.Based on the information that is available from the interviews, can you suggest the next steps policy makers should consider while enabling diffusion of ‘Agrivoltaics’ technology! Can you add a section on this suggesting the possible steps with examples from other technologies that has successfully diffused to the society ?

5.Based on the information that is available from the interviews, what is the best way to diffuse the Technology? Do you observe any parallels on the issues you observe with Agrivoltaics with any other Technology which has succeeded or failed? Please give few examples.

Author Response

1. There are no 'References' given in the article at the end.

Fixed: {Full list of References is provided in text}

 2. Please remove the last line from the Abstract: "It is concluded that building integrated energy and food systems can increase global land productivity, minimize agricultural displacement, and reduce carbon emissions from fossil fuels." I don't think this is the conclusion from this paper.

Fixed: {Removed}

3. It looks like the number of interview samples are very low. In some geographic locations, it has only one participant. There is also one crop farmer. Why not keep the number of samples the same for each profession and geographic region? Then we could have seen differences in perceptions across profession and geographic regions. Do all the participants have an agrivoltaic system or have experience with one?

It is true that the sample size and sampling method do not lend these findings to statistical generalizability or representation of a broader population. However, statistical generalizability is not the aim of interview methodology, which rather seeks theoretical generalizability. Because this study is provisional, external validity of case selection was sought in terms of logical representativeness, diversity, and relevance to the purpose of this study. Rather than increasing the sample size to meet the requirements of statistical generalizability, an explanation is provided (see below) regarding the conventional use of interview samples to generalize theoretical propositions in social science research.

Fixed: {While credible and valuable, samples constructed through non-probability sampling do not lend themselves to generalization [42] nor are the findings generated through interview methodology suitable for statistical generalization or analysis. However, all of the themes discussed as findings were raised by the majority of participants and identify the primary opportunities and barriers to agrivoltaics according to this sample but cannot be quantified or suggested to represent a broader population. Therefore, the findings are not discussed quantitatively to steer clear from suggesting these results are statistically generalizable to the entire agriculture industry.} Line #130-136

Only 4 of the 11 participants have experience with an agrivoltaic system, as denoted by reference to them as “solar grazers” (listed in Table 1). By virtue of snowball sampling, we progressively followed a chain of referrals from study participants to other potential contributors rather than intentionally seeking an equal number of samples from each profession or geographic region. While comparing the differences in perceptions across these characteristics would be stronger if equal representation were captured, it was not exactly the aim of the study to investigate inter-industry variation, but rather to conduct a first survey on industry perspectives in general. The fine-tuned distinctions between actors of various professions/regions within the industry may be considered an area for future research.

Fixed: {Future research should investigate how perceptions vary across geographic regions and agriculture professions (i.e. animal versus crop farming) to study the unique opportunities and barriers for agrivoltaics in the context of local climate and agricultural practice.} Line #449-452

4. Based on the information that is available from the interviews, can you suggest the next steps policy makers should consider while enabling diffusion of ‘Agrivoltaics’ technology!

Fixed: {Future policy research can investigate the role of market mechanisms, such as incentives, in prompting further development of agrivoltaics. Based on these findings, policy makers should consider implementing financial instruments that stimulate both solar and agriculture industry adoption of the technology, while building flexibility into such policies to allow diverse, innovative, and contextually appropriate system designs.} Line #459-469

5. Can you add a section on this suggesting the possible steps with examples from other technologies that has successfully diffused to the society?

Yes, this was covered in great detail in the Diffusion of Innovation work cited in the paper.

Fixed: {To do this, agrivoltaic proponents can model their efforts off of the successful diffusion of wind farm/solar farm integration that focuses on local support [65,66]. Previous research examining diffusion of solar as an innovation among residential adopters highlighted the role of communities of information sharing for promoting adoption [67]; the study presented here is unique in examining the diffusion of the agrivoltaic solar innovation as a community level consideration, but also demonstrates how diffusion of innovation can occur within a social context.} Line 464-469

6. Based on the information that is available from the interviews, what is the best way to diffuse the Technology? Do you observe any parallels on the issues you observe with Agrivoltaics with any other Technology which has succeeded or failed? Please give few examples.

Fixed: {We have outlined what we think are the most important next steps in Section 4.2 and in Table 2. We have added some examples from other areas.}

Reviewer 3 Report

1. Is there any age information about Interview Participant Characteristics in Table 1? If it does, please provide such information. Also, the number of interviewers is insufficient from a statistical perspective, please clarify this.
2. what is the future direction of this study? Will the interview continue to have more interviewers located nationally or in certain regions?

Author Response

  1. Is there any age information about Interview Participant Characteristics in Table 1? If it does, please provide such information.

Age information was not elicited from the study participants.

  1. Also, the number of interviewers is insufficient from a statistical perspective, please clarify this.

Fixed: {While credible and valuable, samples constructed through non-probability sampling do not lend themselves to generalization [42] nor are the findings generated through interview methodology suitable for statistical generalization or analysis. However, all of the themes discussed as findings were raised by the majority of participants and identify the primary opportunities and barriers to agrivoltaics according to this sample but cannot be quantified or suggested to represent a broader population. Therefore, the findings are not discussed quantitatively to steer clear from suggesting these results are statistically generalizable to the entire agriculture industry.} Line #130-136

  1. What is the future direction of this study? Will the interview continue to have more interviewers located nationally or in certain regions?

The future direction of this study does not involve continued interviews but rather an in-depth look at various policy instruments, such as financial incentives, that could potentially make agrivoltaic development more attractive to interested stakeholders and provide market certainty for farmers, as revealed by this study as an important factor in farmer adoption. While international interviewees would be valuable for comparison and insight, it is beyond the scope of this project as we are focused on the opportunities & barriers to agrivoltaic development within the context of the United States regulatory regime.

Round 2

Reviewer 1 Report

The authors made the required changes.

In particular, with the proposed new title (A First Investigation of Agriculture Sector Perspectives on the Opportunities & Barriers for Agrivoltaics), in my opinion the manuscript has all the requirements for publication.